# Three New Species of *Cystolepiota* from Laos and Thailand

Phongeun Sysouphanthong [1,2,3], Naritsada Thongklang [1,2,*], Yuan S. Liu [4] and Else C. Vellinga [5]

1    Center of Excellence in Fungal Research, Mae Fah Luang University, Chiang Rai 57100, Thailand;
     laofungi@gmail.com
2    School of Science, Mae Fah Luang University, Chiang Rai 57100, Thailand
3    Ecology Division, Biotechnology and Ecology Institute, Ministry of Forestry and Agriculture,
     Vientiane P.O. Box 811, Laos
4    Department of Biology, Faculty of Science, Chiang Mai University, Chiang Mai 50200, Thailand;
     yuanshuailiu9@gmail.com
5    UC Herbarium, UC Berkeley, Berkeley, CA 94720-2465, USA; ecvellinga@comcast.net
*    Correspondence: naritsada.t@gmail.com; Tel.: +66-539-16996

**Abstract:** *Cystolepiota* Singer is rarely studied in Southeast Asia; here, we survey and describe three new species of *Cystolepiota* from tropical Laos and Thailand. *Cystolepiota pyramidalis* is related to *C. fumosifolia* (Murrill) Vellinga and *C. pseudofumosifolia* M.L. Xu & R.L. Zhao, but it is distinguished by pale to pastel yellow lamellae. Second, *Cystolepiota thailandica* differs from other members in the genus by the greyish-orange granulose or powdery pileus and stipe covering made up of globose to subglobose and sphaeropedunculate elements. Furthermore, *Cystolepiota rhodella* is characterized by the pink-violet granulose covering of the pileus and stipe and white lamellae with distinctly violet edges. Each species is provided with a full description of the morphological characters, photos in situ, line drawings of the microcharacters, discussion of related and similar species, and molecular data.

**Keywords:** Agaricaceae; diversity; lepiotaceous fungi; new species; phylogeny; Southeast Asia; taxonomy



## 1. Introduction

The genus *Cystolepiota* Singer (Agaricaceae s.l.) is very diverse, and all species are saprotrophic. The estimates of the number of species in the genus range from twelve [1] to forty-five [2]. The species are characterized by small pluteoid or lepiotoid basidiomata with a granulose to powdery covering, a white to pale lilac spore print, and the hyaline, non-dextrinoid, or dextrinoid spores that can be smooth or slightly roughened, with or without cheilocystidia and pleurocystidia, a regular hymenophoral trama, the epitheloid pileus and stipe covering made up of globose, broadly ellipsoid to elongated elements, and the absence or presence of clamp connections. The species with elongated elements in the pileus covering and bi-nucleate spores were placed in the separate genus *Pulverolepiota* [3]; this genus was later considered a section of *Cystolepiota*. The three sections in the genus, viz., sect. *Cystolepiota* Singer, sect. *Pulverolepiota* (M. Bon) Vellinga, and sect. *Pseudoamyloideae* Singer & Clém. are distinguished based on morphological characters such as the shape of the pileus covering elements (elongated and irregular in sect. *Pulverolepiota*) and the reaction of the spore wall in Melzer's reagent (dextrinoid in sect. *Pseudoamyloideae*) [4]. Species in sect. *Pulverolepiota* also differ in the binucleate spores, whereas the other species have uninucleate spores.

*Melanophyllum* Velen. species share many characters with *Cystolepiota* but differ in the colored, ornamented spores. Molecular phylogenetic studies [5,6] showed that *Melanophyllum* and *Cystolepiota* together form a monophyletic group; *Melanophyllum* is the oldest name available for this group, but there are no sequence data available for its type, *M. canali* Velen.; *Fusispora* Fay., typified by species *L. sistrata* Quél. Fr., which is generally considered

to belong to *Cystolepiota*, is considered a nomen dubium by Donk [7], as the spore sizes given are about 16 × 5 μm [8], a size not found in any *Cystolepiota* species.

Some authors, e.g., Knudsen [9] and Bon [10] considered *Lepiota* sect. *Echinatae*, also characterized by globose to ellipsoid elements in the pileus covering, part of *Cystolepiota*; in 1980, Knudsen [9] came back from that idea and placed those species again in *Lepiota*, but Bon [11] erected a separate genus, *Echinoderma* (Locq. ex Bon) Bon, for those species. Recently, Vellinga [12] and Hou and Ge [6] showed that, based on molecular characters, those *Echinoderma* species with small ellipsoid spores still belong in *Lepiota* and that the species, such as *E. asperum* (Pers.) Bon, with elongated spores are separate from both *Lepiota* and *Cystolepiota* and are placed in *Echinoderma*.

*Cystolepiota* is rarely studied in Laos and Thailand. Only *Cystolepiota seminuda* (Lasch) Bon and *Cystolepiota sistrata* (Fr.) Singer ex Bon and Bellù were recorded in Thailand [13]. Sysouphanthong et al. [14] reported *C.* aff. *icterina* F.H. Møller ex Knudsen, and *C. pulverulenta* (Huijsman) Vellinga from Laos. However, the records of those species were only based on their morphology. In this study, three new species are examined from Laos and Thailand; details of their morphology, ecology, and distribution and a molecular phylogenetic analysis are presented.

## 2. Materials and Methods

### 2.1. Collecting and Material Examination

During the rainy seasons of 2012–2018, samples were collected in Chiang Mai and Chiang Rai Provinces (Northern Thailand) and Oudomxay Province (Northern Laos). Thai specimens were deposited in the Herbarium of Mae Fah Luang University (MFLU), and Lao specimens were deposited in the National Herbarium of Laos (HNL). Fresh samples were photographed in situ, and the ecology of the original places was recorded. After collection, macro- and microscopic characters of the specimens were observed in the laboratory. The main features of fresh samples are pileus, lamellae, stipe, annulus, context, spore print, taste, and odor. Color features were described and followed the codes of Kornerup and Wanscher [15]. After macroscopic observation, specimens were dried in a hot air dryer around 30–40 °C for 24 h and deposited to preserve in the fungarium for future studies. Microscopical characteristics were observed and illustrated from fresh or dry specimens with a compound microscope. Distilled water and 3–10% KOH were used to observe color features; Melzer's reagent, cotton blue, and cresyl blue were used to examine spore reaction, and ammoniacal Congo red was used to stain spore walls and hyphae. Spore measurements were made from 25 spores of each basidioma of each collection. The terminology of the features followed Vellinga and Noordeloos [4]. The following abbreviations are used: "avl" for average length, "avw" for average width, "Q" for quotient of length and width, and "avQ" for average quotient.

### 2.2. Phylogenetic Study

DNA was extracted from dried collections according to the instructions of the Biospin Fungus Genomic DNA Extraction Kit (Bioer Technology Co., Ltd., Hangzhou, China). For the PCR and PCR amplification, primers ITS1/ITS4 or ITS1-F/ITS4 were used for the nrITS1, 5.8S, and nrITS2 regions [16,17], primers LR0R/LR5 for the large subunit region (LSU) [18,19], and primers fRPB2-5C/fRPB2-7CR for the polymerase II second largest subunit (RPB2) region [20]. Sequencing was performed by Shanghai Majorbio Bio-Pharm Technology Co., Ltd. The sequences were checked and assembled using the SeqMan program (DNAStar, Madison, WI, USA), and new sequences were deposited in GenBank (https://www.ncbi.nlm.nih.gov/genbank/ (accessed on 1 June 2022)).

Available sequences of *Cystolepiota*, representatives of *Lepiota*, and of the closely related genera *Echinoderma*, *Melanophyllum*, and *Smithiomyces* were obtained from GenBank. Each dataset was first aligned using MAFFT version 7.130-win32 [21,22]. The final dataset comprised 82 collections and 2411 characters (including gaps), which were 78 collections and 751 characters from ITS, 51 collections and 881 characters from LSU, and 35 collections

and 779 characters for RPB2. The final alignments were submitted to TreeBASE (ID: 29388). A maximum-likelihood (ML) analysis was performed using RAxML version 7.2.6 [23] with GTRGAMMAI as the model of evolution, and branch support was estimated over 1000 bootstrap partitions (BP) with the rapid bootstrap option. A Bayesian inference (BI) analysis was performed with MrBayes 3.1.2 [24]. The best substitution model of individual genes was determined using MrModelTest v.2.3 [25]. The best selected model (GTR+I+G) was for ITS, LSU, and RPB2. For the BI analysis setting, three Metropolis-coupled Markov chain Monte Carlo (MCMC) runs, each with four heated chains and two cold chains, and the run was conducted for 5 million generations and sampled every 1000 generations, with the first 10% discarded as burn-in. The phylogram results of all analyses were exported and edited in TreeView 1.0.0.0 [26]. The phylograms were edited in the software of Adobe Illustrator CS3.

## 3. Results and Discussion

### 3.1. Phylogeny

The maximum-likelihood phylogram of representative lepiotoid mushrooms was based on a multi-gene DNA dataset made up of ITS, LSU, and RPB2 gene regions (Table 1). The alignment comprised 74 specimens with 2103 characters in total (including the gaps). The best RaxML phylogram, with a final likelihood value of −23,323.039705, is presented. The matrix had 1151 distinct alignment patterns with 38.63% undetermined characters or gaps. The estimated base frequencies were as follows: A = 0.253717, C = 0.213111, G = 0.263577, and T = 0.269595; substitution rates, AC = 1.484055, AG = 4.134055, AT = 1.770416, CG = 0.419899, CT = 6.554890, and GT = 1.000000; gamma distribution shape parameter, $\alpha$ = 0.929050. The phylogram topology derived from the Bayesian analysis was similar to that derived from the ML analysis. Bootstrap values of ML $\geq$ 70% and bootstrap values of BI $\geq$ 0.95 are indicated in Figure 1.

**Table 1.** GenBank accession numbers, geographical origins, and voucher numbers of taxa used for the phylogenetic analysis.

| Taxon | Country | Voucher Number | GenBank Accession Number | | |
|---|---|---|---|---|---|
| | | | ITS | LSU | RPB2 |
| *Agaricus campestris* | China | LAPAG370 | KM657927 | KR006607 | KT951556 |
| *Agaricus friesianus* | China | ZRL2012601 | KX657026 | KX656970 | KX685048 |
| *Cystolepiota adulterina* | Italy | 475 | JF907978 | - | - |
| *Cystolepiota bucknallii* | The Netherlands | ecv1761 | AY176458 | - | - |
| *Cystolepiota bucknallii* | Italy | 490 | JF907979 | - | - |
| *Cystolepiota cystidiosa* | USA | MICH18884 | U85333 | U85298 | - |
| *Cystolepiota cystophora* | Costa Rica | DUKE-JJ87 | U85332 | U85297 | - |
| *Cystolepiota fumosifolia* | USA | ecv3278 | EF121817 | - | - |
| *Cystolepiota hetieri* | The Netherlands | ecv2237 | AY176459 | - | - |
| *Cystolepiota hetieri* | Italy | 782 | JF907982 | - | - |
| *Cystolepiota hetieri* | China | 420526MF0093 | MG694259 | - | - |
| *Cystolepiota hetieri* | Canada | HRL0772 | MH979434 | - | - |
| *Cystolepiota hetieri* | Canada | HRL1277 | MH979438 | - | - |
| *Cystolepiota hetieri* | USA | HRL2162 | MH979463 | - | - |
| *Cystolepiota hetieri* | China | HKAS 84189 | MN810139 | MN810094 | MN820976 |
| *Cystolepiota hetieri* | China | HKAS 53554 | MN810143 | MN810102 | MN820977 |
| *Cystolepiota luteohemisphaerica* | Ecuador | TL_11724 | AM946477 | AM946476 | - |
| *Cystolepiota pseudofumosifolia* | China | HKAS 104303 | MN810150 | MN810095 | MN820973 |
| *Cystolepiota pseudofumosifolia* | China | HKAS 105918 | MN810152 | MN810108 | MN820974 |
| *Cystolepiota pulverulenta* | USA | ecv1872 | AF391036 | AY176349 | - |
| *Cystolepiota pulverulenta* | USA | ecv1763 | AF391037 | - | - |
| ***Cystolepiota pyramidalis*** | **Laos** | **HNL502500** | **MZ574554** | **MZ569511** | **-** |
| ***Cystolepiota pyramidalis*** | **Thailand** | **MFLU 12-1774** | **MZ574555** | **MZ569512** | **-** |

**Table 1.** *Cont.*

| Taxon | Country | Voucher Number | GenBank Accession Number | | |
|---|---|---|---|---|---|
| | | | ITS | LSU | RPB2 |
| *Cystolepiota rosea* | Italy | 781 | JF907981 | - | - |
| *Cystolepiota seminuda* | The Netherlands | H.A. Huijser s.n. | AY176350 | AY176351 | - |
| *Cystolepiota seminuda* | Italy | 9247 | JF907983 | - | - |
| *Cystolepiota seminuda* | USA | Smith 2018 | MK573889 | - | - |
| *Cystolepiota* aff. *seminuda* | China | HKAS 73969 | MN810144 | MN810100 | MN820979 |
| *Cystolepiota* aff. *seminuda* | China | HKAS 92275 | MN810149 | MN810101 | MN820980 |
| *Cystolepiota sistrata* | Canada | HRL1282 | MH979429 | - | - |
| *Cystolepiota sistrata* | Italy | 491 | JF907980 | - | - |
| *Cystolepiota sistrata* | USA | HRL2161 | MH979462 | - | - |
| *Cystolepiota* sp. | China | ZRL2011054 | KF804000 | - | - |
| *Cystolepiota* sp. | China | ZRL2012038 | KF804001 | - | - |
| *Cystolepiota* sp. | Canada | UBC_F24523 | MF955171 | - | - |
| *Cystolepiota* sp. | USA | HRL1900 | MH979456 | - | - |
| *Cystolepiota* sp. | USA | MycoMap_7788 | MK560110 | - | - |
| *Cystolepiota* sp. | USA | MycoMap_7792 | MK560111 | - | - |
| *Cystolepiota* sp. | China | HKAS 78850 | MN810142 | MN810103 | MN820978 |
| *Cystolepiota* sp. | China | HKAS 105719 | MN810151 | MN810109 | MN820975 |
| *Cystolepiota* sp. *(Echinoderma)* | Thailand | ecv3896 | - | HM488789 | - |
| *Cystolepiota* sp. *(Echinoderma)* | Malaysia | LAM 0001 | - | KY090841 | - |
| *Cystolepiota* sp. | India | HATFD14_95 | KU847887 | - | - |
| **Cystolepiota thailandica** | **Thailand** | **MFLU 22-0017** | **MZ574556** | **MZ569513** | **-** |
| **Cystolepiota rhodella** | **Laos** | **HNL501799** | **MZ574551** | **MZ569508** | **MZ508496** |
| **Cystolepiota rhodella** | **Thailand** | **MFLU 22-0019** | **MZ574552** | **MZ569509** | **MZ574090** |
| **Cystolepiota rhodella** | **Thailand** | **MFLU 09-0050** | **MZ574553** | **MZ569510** | **-** |
| *Echinoderma asperum* | North Macedonia | HKAS 106783 | MN810133 | MN810088 | MN820967 |
| *Echinoderma asperum* | USA | HKAS 84214 | MN810135 | MN810089 | MN820964 |
| *Echinoderma asperum* | USA | HKAS 84240 | MN810136 | MN810090 | MN820965 |
| *Echinoderma asperum* | China | HKAS 106782 | MN810134 | MN810087 | MN820962 |
| *Echinoderma asperum* | China | HKAS 105694 | MN810153 | MN810106 | MN820966 |
| *Echinoderma asperum* | China | HKAS 77440 | MN810145 | MN810096 | MN820963 |
| *Echinoderma asperum* | India | NEHU.MBSRJ.55 | KP843884 | MG253012 | - |
| *Echinoderma flavidoasperum* | China | HKAS 87905 | MN810147 | MN810098 | MN820969 |
| *Echinoderma flavidoasperum* | China | HKAS 76527 | MN810146 | MN810097 | MN820968 |
| *Echinoderma hystrix* | France | H.A. Huijser | AY176377 | AY176378 | - |
| *Echinoderma* sp. | China | HKAS 70488 | MN810148 | MN810099 | MN820970 |
| *Echinoderma* sp. | China | HKAS 106735 | MN810154 | MN810107 | MN820971 |
| *Lepiota* aff. *carinii* | Hungary | NL-2202 | - | MK277953 | - |
| *Lepiota alba* | China | HKAS 90371 | MN810115 | MN810075 | MN820946 |
| *Lepiota asperula* | Canada | S.D.Russell HRL1281 | MH979440 | - | - |
| *Lepiota castanea* | China | HKAS.84179 | MN810119 | MN810077 | MN820960 |
| *Lepiota clypeolaria* | China | HKAS 87248 | MN810123 | MN810080 | MN820941 |
| *Lepiota echinacea* | China | HKAS 105582 | MN810155 | MN810104 | MN820954 |
| *Lepiota geocarpa* | USA | UTC00143916 | HQ020412 | EU130550 | MN820945 |
| *Lepiota geophana* | USA | UTC00253060 | HQ020411 | HQ020421 | MN820944 |
| *Lepiota jacobi* | China | HKAS 48802 | MN810138 | GU199356 | MN820953 |
| *Lepiota magnispora* | China | HKAS 61622 | JN944089 | JN940285 | JN993693 |
| *Lepiota omninoflava* | China | HKAS 106734 | MN810157 | MN810092 | MN820951 |
| *Melanophyllum eyrei* | South Korea | ASIS23988 | KF953546 | - | - |
| *Melanophyllum eyrei* | Sweden | TL6692 | AY176493 | - | - |
| *Melanophyllum haematospermum* | USA | ecv2517 | AF391039 | AY176456 | - |
| *Melanophyllum haematospermum* | The Netherlands | ecv2249 | AF391038 | AY176455 | - |
| *Melanophyllum haematospermum* | Canada | S.D.Russell HRL1807 | MH979452 | - | - |
| *Melanophyllum haematospermum* | Italy | 913 | JF908498 | - | - |

**Table 1.** *Cont.*

| Taxon | Country | Voucher Number | GenBank Accession Number | | |
|---|---|---|---|---|---|
| | | | ITS | LSU | RPB2 |
| *Smithiomyces asiaticus* | China | HKAS 84395 | MW522986 | MW716269 | MW736566 |
| *Smithiomyces dominicanus* | Dominican Republic | BSD126144 | KR604686 | MW716266 | MW736563 |
| *Smithiomyces heterosporus* | China | HKAS 84392 | MW522985 | MW716268 | MW736565 |
| *Smithiomyces lepiotoides* | China | HKAS 54390 | MW522984 | MW716270 | - |
| *Smithiomyces mexicanus* | Switzerland | UTC259587 | MW723225 | MW716267 | MW736564 |

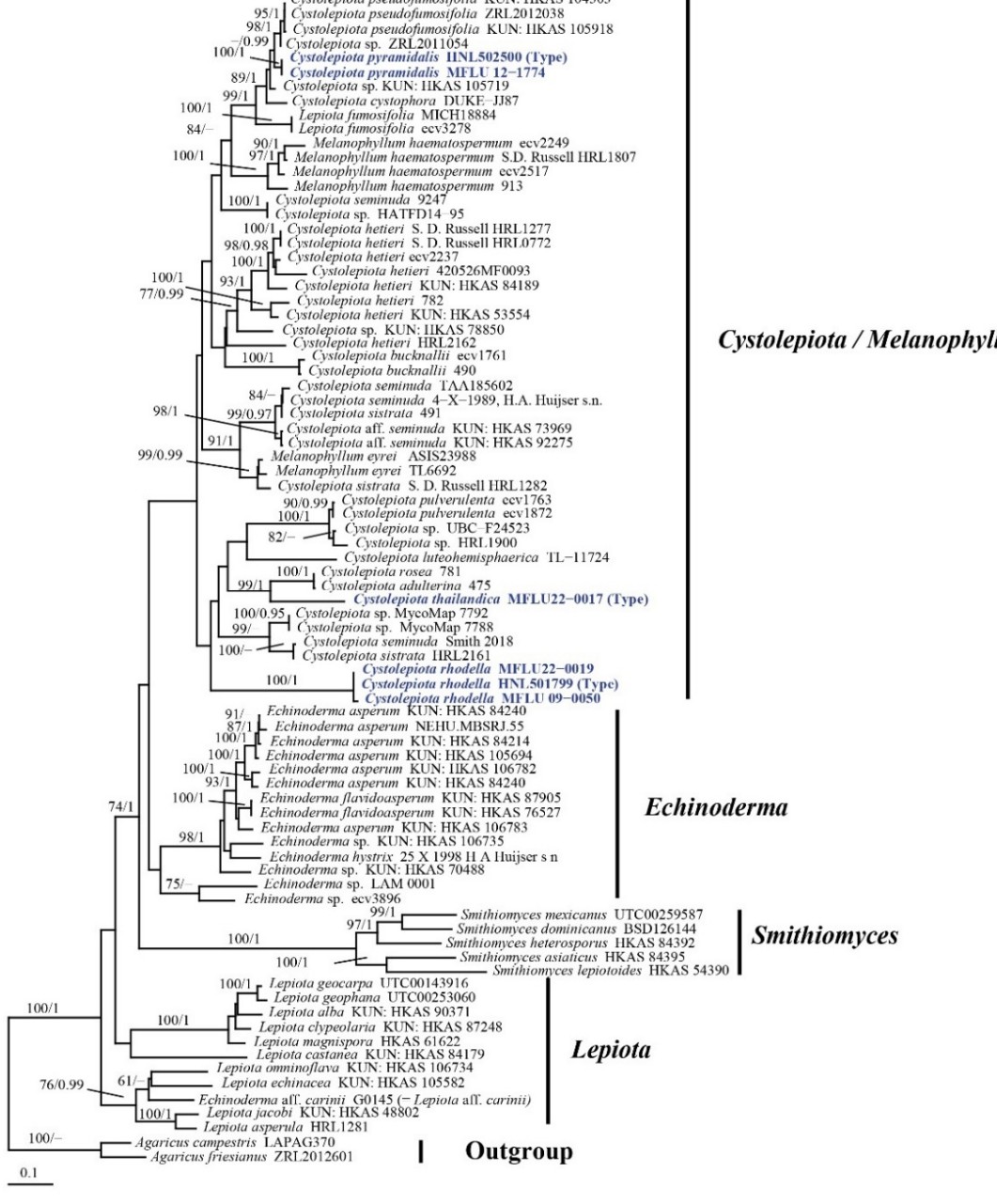

**Figure 1.** Maximum-likelihood phylogram of *Cystolepiota* and *Melanophyllum* specimens based on combined nrITS-LSU and RPB2 sequences. New sequences generated for this study are in blue. Bootstrap values of ML ≥ 60%, and BI ≥ 0.95 are indicated above the branches (ML/BI). *Agaricus campestris* L. and *Agaricus friesianus* L.A. Parra, Olariaga & Callac are chosen as outgroup.

The maximum-likelihood phylogram (Figure 1) shows three major clades: a clade of *Cystolepiota* and *Melanophyllum* species, *Echinoderma*, and a clade of *Lepiota* species.

The phylogram in Figure 1 shows clearly that much more work on the genus and its species has to be conducted; many species concepts (e.g., of *C. seminuda* and *C. hetieri*) are not settled yet. A second conclusion is that the morphology-based sections are not reflected in the tree based on molecular characters. The three Thai/Laotian species fall in different clades, with *C. pyramidalis* close to *C. pseudofumosifolia* and *C. fumosifolia*; *C. rhodella* takes a rather isolated position on a long branch basal to the *C. hetieri* complex. The third species, *C. thailandica*, is closely related to species in the *C. seminuda* complex.

### 3.2. Taxonomy

3.2.1. *Cystolepiota pyramidalis* Sysoup. and Thongkl. sp. nov.

MycoBank: MB843144; FoF number: FoF 10598; Figures 2 and 3.

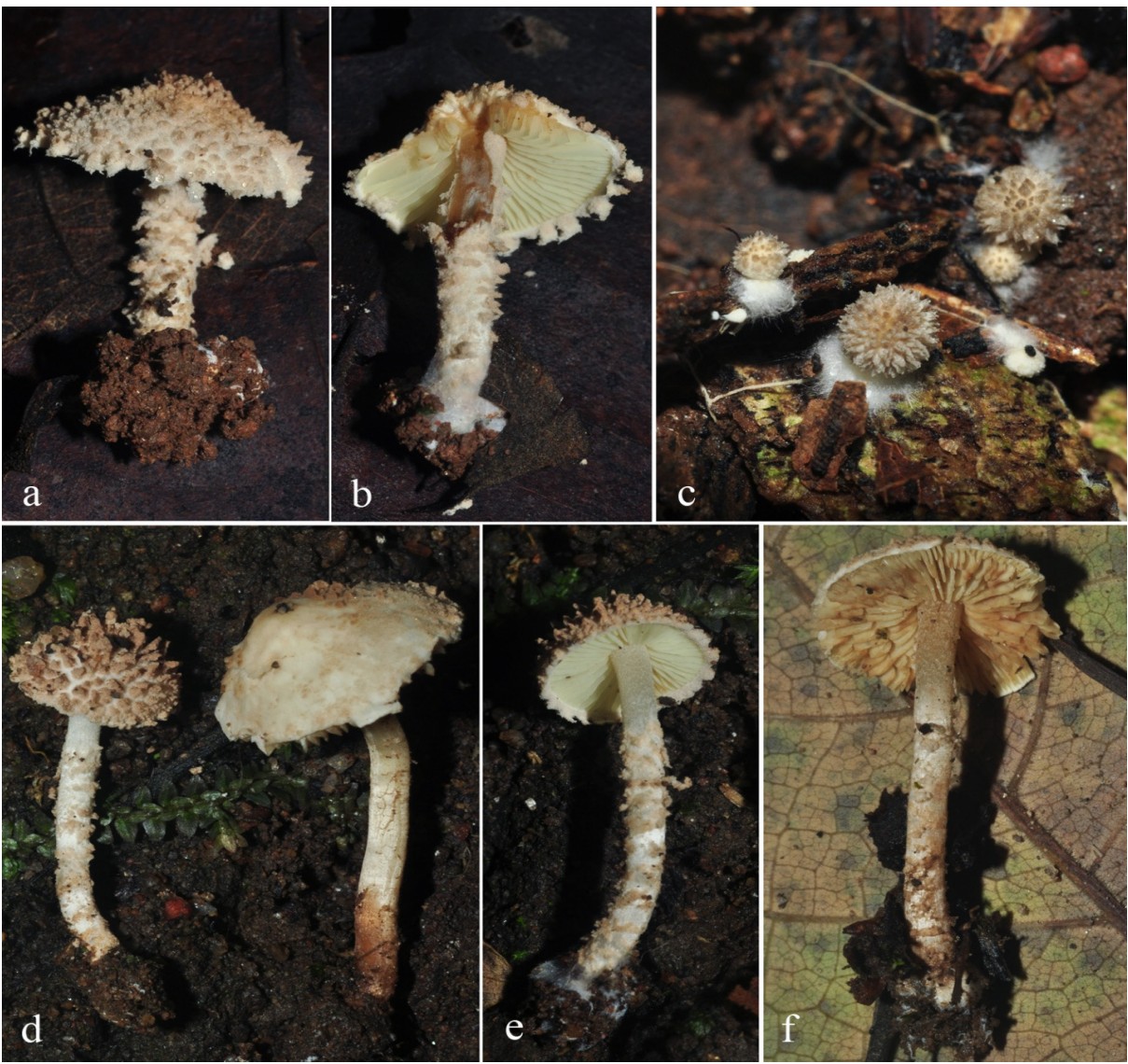

**Figure 2.** Macrocharacters of *Cystolepiota pyramidalis*. (**a–c**) = MFLU12-1774 (paratype), (**d–f**) = HNL 502500 (holotype).

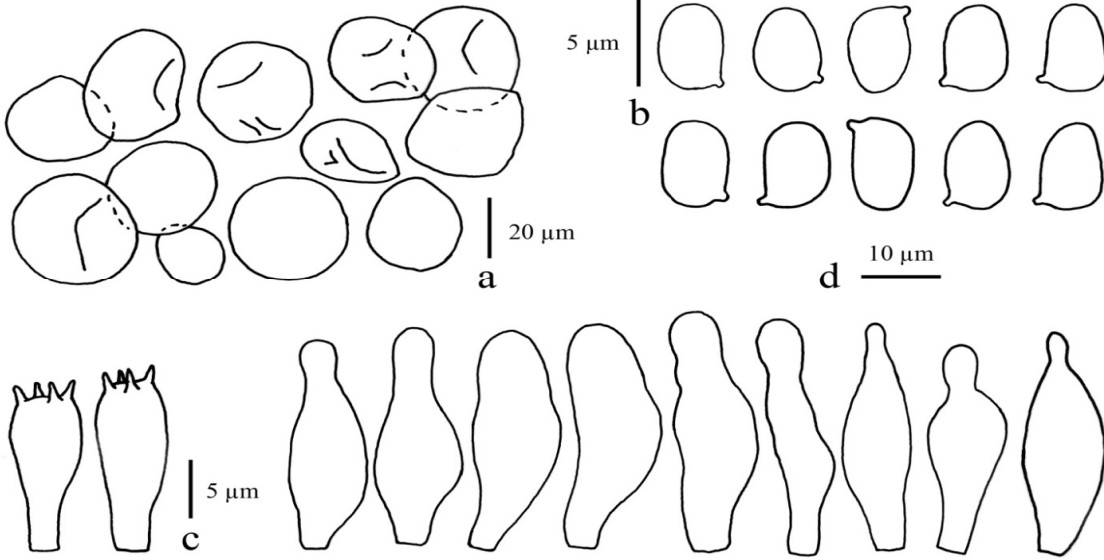

**Figure 3.** Microcharacters of *Cystolepiota pyramidalis* (HNL502500, holotype). (**a**) = elements of pileus covering, (**b**) = basidiospores, (**c**) = basidia, (**d**) = cheilocystidia.

Etymology—the name "*pyramidalis*" refers to the pyramidal shape of the squamules on the pileus.

Diagnosis—*C. pyramidalis* is recognized by basidiomata covered with light brown to brown pyramidal or irregular pyramidal squamules, pale yellow lamellae, hyaline and ellipsoid-ovoid, smooth basidiospores, variably shaped cheilocystidia with or without excrescence at the apex, the absence of pleurocystidia, epitheliod pileus and stipe covering, and the presence of clamp connections.

Holotype—Laos, Oudomxay Province, Xay District, Houay Houm Village, 18 August 2014, P. Sysouphanthong, PS2014-8290 (HNL502500).

Pileus 15–45 mm diam., first parabolic or campanulate, expanding to convex or umbonate, with straight margin, when young completely covered with crowded pyramidal or irregular pyramidal squamules, brownish (6D7–8), soon breaking up into brownish (6D7–8) to light brown (6D4–5) pyramidal to granular pyramidal squamules or warts toward margin, sparse or fragile when mature, on orange-white to pale orange (5A2–3) background; margin covered with concolorous pyramidal to granular pyramidal velar remnants. Lamellae free, 3–5 mm wide, pale yellow (3A3), becoming brownish orange (6C4–6) when touched or mature, broadly ventricose, with 3–4 lamellulae, with concolorous smooth to slightly eroded edge. Stipe 25–40 × 4–5.5 mm, cylindrical, covered with concolorous squamules to those on pileus, sparse at apex, fragile when mature, on orange-white to pale orange (5A2–3) background. Annulus an annular zone with velar remnants and concolorous to pileus margin, sometimes fragile with age. Context white in pileus, up to 1 mm thick at center; hollow in stipe and concolorous with surface. Odor and taste not observed. Spore pint white.

Basidiospores (50,2,2) 3.8–4.5 × 2.5–3.2 μm, avl × avw = 4.1 × 3.0 μm, Q = 1.25–1.6, avQ = 1.37, ellipsoid-ovoid in frontal view, ellipsoid in side-view, slightly thick-walled, smooth, hyaline, non-dextrinoid, non-amyloid, cyanophilous. Basidia 15–18 × 4.5–7 μm, clavate, thin-walled, hyaline, four-spored, sometimes two-spored. Cheilocystidia 20–40 × 7–15 μm, variable in shape, irregular cylindrical, fusiform, narrowly utriform to utriform, lageniform, clavate with a narrowed apex, slightly thick-walled, hyaline. Pleurocystidia absent. Pileus covering an irregular epithelium composed of globose to subglobose elements, 35–65 μm in diam., slightly thick-walled, smooth, with pale brown to brown parietal and intracellular pigments. Stipe covering an irregular epithelium same as on pileus. Clamp connections present.

Habitat and habit—growing in small groups, saprotrophic on humus-rich soil of mixed deciduous forest with *Castanopsis* spp. and *Lithocarpus* spp. dominant; the species is rare and so far known from Oudomxay province, northern Laos, and Chiang Rai province, northern Thailand.

Additional material examined—Thailand, Chiang Rai Province, Muang District, Phoo Kham Fah Village, 15 August 2012, P. Sysouphanthong, PS2012-11 (MFLU12-1774, paratype).

Notes—*Cystolepiota pyramidalis* is rare in Laos and Thailand; it was found in two locations, not far apart from each other at similar elevations, viz., Oudomxay Province of northern Laos and Chiang Rai Province of northern Thailand. The specimens from these two locations are identical both in morphological and in molecular characters. It is distinguished by light brown to brown pyramidal squamules on basidiomata and pale yellow lamellae. This new species is distinguished from other species by yellow lamellae and the distinct pyramidal shape of the squamules on the pileus.

*Cystolepiota pyramidalis* belongs to a clade of similar species (Figure 1). *C. fumosifolia*, known from North America and Europe (as *C. cystidiosa* (A.H. Smith) Bon, *C. luteicystidiata* (D.A. Reid) Bon, and *L. lycoperdoides* Kreisel), also has pyramidal granular warts on the pileus, but the cheilocystidia and abundant pleurocystidia have yellow contents, and the cheilocystidia are covered with yellow exudate. *C. pseudofumosifolia* from China has white lamellae and lacks the pyramidal warts on the cap. It lacks pleurocystidia, just like *C. pyramidalis*.

There are some other species with yellow lamellae in the genus. First, *Cystolepiota bucknallii* (Berk. and Broome) Singer & Clémençon, known from temperate regions in Europe and North America, has pale yellow to pastel yellow lamellae, but the pileus and stipe are covered with lilac granulose squamules, and the much longer (7–9 μm) spores are dextrinoid [27,28]. Second, *Cystolepiota seminuda* (Lasch) M. Bon also has white to yellowish creamy lamellae with a pale lemon-yellow tinge, but its basidiomata are much smaller and have a white to cream densely floccose-verrucose covering on the pileus, and it lacks cheilocystidia [28,29].

Additionally, *Cystolepiota oliveirae* P. Roux, M. Paraíso, J.-P. Maurice, A.-C. Normand & F. Fouchier, described from Portugal, has distinctly white to reddish-brown squamules or warts on the pileus and stipe, but the species has white to cream lamellae and rough basidiospores [30], while *C. pyramidalis* has yellow lamellae and smooth basidiospores.

### 3.2.2. *Cystolepiota thailandica* Yuan S. Liu, Sysouph. and Thongkl. sp. nov.

MycoBank: MB843145; FoF number: FoF 10597; Figures 4 and 5.

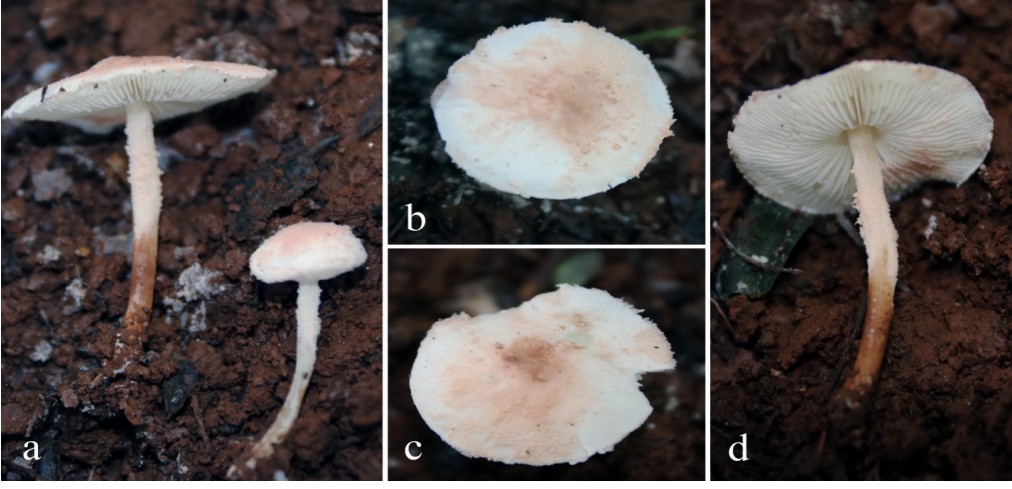

**Figure 4.** Macrocharacters of *Cystolepiota thailandica*. (**a**,**b**) = MFLU22-0017 (holotype), (**c**,**d**) = MFLU22-0018 (paratype).

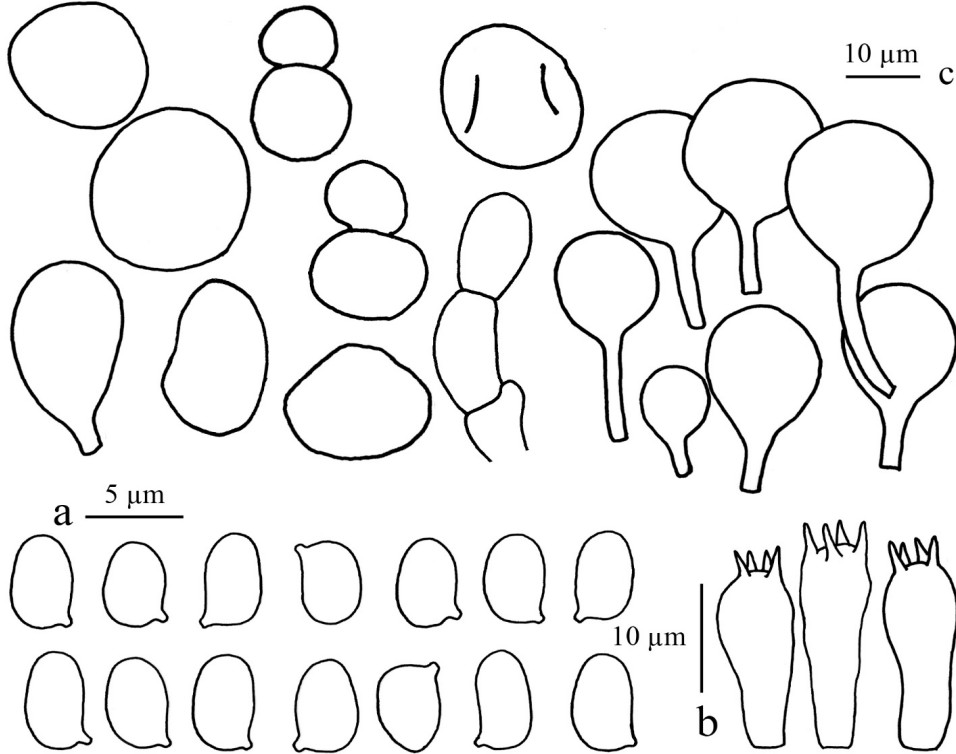

**Figure 5.** Microcharacters of *Cystolepiota thailandica* (MFLU22-0017, holotype). (**a**) = basidiospores, (**b**) = basidia, (**c**) = pileus covering cells.

Etymology –the species epithet 'thailandica' is derived from the name of the country where the material was collected.

Diagnosis—*Cystolepiota thailandica* has basidiomata that are covered with greyish-orange granules or powder, white lamellae, hyaline and ellipsoid basidiospores, clavate basidia, no cheilocystidia or pleurocystidia, a pileus and stipe covering made up of globose to subglobose and sphaeropedunculate elements and clamp connections.

Holotype—Thailand, Chiang Rai Province, Muang District, Mae Fah Luang University Campus, 11 January 2019, Yuan S. Liu, STO-2019-062 (MFLU22-0017).

Pileus 25–32 mm diam., first hemispherical to convex, expanding to convex or slightly plano-concave, with inflexed margin, when young granulose to powdery, greyish orange (6B3–5), soon breaking up, granulose to powdery at center toward margin, sometimes fragile when mature, on white powdery background; margin covered with granulose to powdery velar remnants, concolorous to those on surface. Lamellae free, 1.5–2 mm wide, crowded, white, broadly ventricose, with 3–4 lamellulae, with concolorous eroded edge, turning light brown to brown (7D5–8) when touched. Stipe 25–35 × 1.5–2.5 mm, cylindrical, completely covered with white to greyish orange (6B3–5) granulose to powdery velar remnants, on white background, turning light brown to brown (7D5-8) when touched. Annulus an annular zone with white to greyish-orange (6B3–5) remnants. Context white in pileus, 1–1.5 mm thick at center; hollow in stipe and concolorous with surface. Odor and taste not observed. Spore print white.

Basidiospores (50,2,2) 4.0–5.0 × 1.8–2.5 μm, avl × avw = 4.30 × 2.24 μm, Q = 1.7–2.0, avQ = 1.91, ellipsoid in frontal view, ellipsoid in side-view, slightly thick-walled, smooth, hyaline, non-dextrinoid, non-amyloid. Basidia 15–20 × 5.0–7.0 μm, short clavate to clavate, thin-walled, hyaline, four-spored. Cheilocystidia absent. Pleurocystidia absent. Pileus covering an epithelium composed of globose to subglobose and sphaeropedunculate element cells, 10–35 μm wide, hyaline to pale brown parietal pigments. Stipe covering an epithelium similar to that on pileus. Clamp connections present in all tissues.

Habitat and habit—growing solitary or in a small group with few basidiomata, saprotrophic on humus-rich soil with dead leaves. Found under cultivated trees of *Ficus annulata* Blume.

Additional materials examined—Thailand, Chiang Rai Province, Muang District, Mae Fah Luang University Campus, 11 January 2019, Yuan S. Liu, STO-2019-063 (MFLU22-0018).

Notes—*Cystolepiota thailandica* is not related to any species in the genus by morphology. *Cystolepiota seminuda* (Lasch) Bon is similar to *C. thailandica* by having globose and sphaeropedunculate element cells on pileus and stipe covering and an absence of cheilo- and pleurocystidia but is different in having a white to pale pink or yellowish pileus and stipe covering and slightly larger basidiospores (3.5–5.5 × 2.0–3.0 μm) [29]. *Cystolepiota bucknallii* (Berk. and Broome) Singer & Clémençon is different by having a lilac or violaceous powdery covering on the pileus and stipe, pale yellow to pastel yellow lamellae, and lacking sphaeropedunculate element cells [27,28]. *Cystolepiota pseudogranulosa* (Berk. & Broome) Pegler, described from Sri Lanka, differs in the strongly dextrinoid spores and the irregularly shaped pileus covering elements [31].

The phylogram shows that *C. thailandica* is separated from the other species for which sequence data are available (Figure 1).

### 3.2.3. *Cystolepiota rhodella* Sysoup. and Thongkl., sp. nov.

MycoBank: MB843146; FoF number: FoF 10599; Figures 6 and 7.

Etymology—the name "rhodella" is from the pinkish-ruby-brown color of the basidiomata.

Diagnosis—*Cystolepiota rhodella* has basidiomata covered with violet-brown to greyish-ruby flocculose squamules, white lamellae with violet-brown edge, oblong- ovoid basidiospores, clavate basidia, no pleurocystidia, abundant moniliform to flexuous cheilocystidia with a long apical excrescence and containing pale yellow mucilaginous contents, pileus and stipe covering epithelioid, made up of globose to subglobose cells and clamp connections.

Holotype—Lao PDR, Oudomxay Province, Xay District, Houay Houm Village, 20°31′11″ N, 101°53′27″ E, alt. 985–940 m, 12 September 2014, P. Sysouphanthong, P1484 (HNL501799).

Pileus 10–32 mm diam., first conical to paraboloid, expanding to campanulate or convex, often umbonate with broad umbo, with straight to incurved margin, completely covered by flocculose squamules when young, violet-brown (10E4–8 and 11E6–8) to greyish ruby or ruby (12C5–8, 12D5–8, and 12E5–8), darker at center, on drying becoming darker, greyish brown to dark brown (8F3–4), on white to pinkish-white or pale red (8A2–3) background; margin with flocculose squamules, concolorous with those on surface, often with white to orange grey (5B2) cortina connecting with stipe when young. Lamellae free, white to whitish, 2–4 mm wide, broadly ventricose, with wavy eroded edge, concolorous with squamules on pileus and stipe, with three lamellulae. Stipe 25–40 × 2–5 mm, cylindrical, covered with crowded flocculose squamules, concolorous with those on pileus, on white to pinkish white or pale red (8A2–3) background, with white rhizomorphs at base; hollow. Annulus an annular zone with concolorous flocculose squamules at upper part of stipe, with concolorous cortina as on pileus margin. Context white in pileus and up to 2 mm thick at umbo, white in stipe. Odor and smell not observed. Spore print white to whitish.

Basidiospores (50,2,2) 3.8–4.2 × 2–3 μm, avl × avw = 4 × 2.4 μm, Q = 1.4–1.9, avQ = 1.7, oblong-ovoid in side view, some with straight base, and with rounded or slightly acute apex, ellipsoid to oblong in frontal view, hyaline, thick-walled, dextrinoid, non-amyloid. Basidia 13–15 × 4.5–6 μm, clavate, hyaline, slightly thick-walled, four-spored. Lamella edge sterile, with abundant cheilocystidia. Cheilocystidia 35–40 × 4–7 μm, moniliform to flexuous with long appendiculate apex, some narrowly lageniform, with pale yellow mucilaginous contents, hyaline and slightly thick-walled. Pleurocystidia absent. Pileus covering an irregular epithelium composed of globose (12–65 μm diam.) to subglobose (17–23 × 12–18 μm) elements in upper layers, with oblong (25–38 × 15–20 μm) elements in lower layers, thin-walled, with pale brown parietal and intracellular pigments. Stipe covering in the squamules an irregular epithelium same as in pileus covering. Clamp connections present.

Habitat and habit—growing solitary or in a small group, saprotrophic on humus-rich soil with dead leaves, in various habitats, e.g., mixed deciduous forest with *Castanopsis* spp. and *Lithocarpus* spp. as dominant tree species in northern Laos, in dipterocarp forest with *Shorea obtusa* Wall. ex Blume dominant, and in deciduous forest of *Ficus annulata* Blume in northern Thailand.

Materials examined: Thailand, Chiang Rai Province, Pa Daed District, Pha Ngae Sub-district, 19°34′57″ N, 100°00′51″ E, alt. 510–540 m, 28 August 2018, P. Sysouphan-thong, PS2018-138 (MFLU22-0019); Chiang Mai, Mae Taeng District, Pong Duad Village, 16°06′16.1″ N, 99°43′07.9″ E, alt. 780–805 m, 9 August 2007, P. Sysouphanthong, PNG31 (MFLU09-0050).

Note—*Cystolepiota rhodella* was found in dipterocarp forests (and other habitats) in Northern Laos and Northern Thailand. It is easily recognized by violet-brown or greyish-ruby to ruby squamules on basidiomata and white lamellae with violet-brown edges. Because of its unique color and the colored lamella edge, the species stands out, and there are few similar species. *Melanophyllum haematospermum* (Bull.) Kreisel differs from *C. rhodella* by cinnamon or pale brownish-gray to light grayish-brown basidiomata and cinnamon-red to dark brown lamellae [32]. Microscopically it differs in the subtly ornamented colored spores and cystidia that are in general absent or, if present, inconspicuous; the exsiccata turn black [33]. The phylogram (Figure 1) showed that *C. rhodella* is separated from other species; it is basal to the clade made up of specimens of *C. hetieri* (Boud.) Singer.

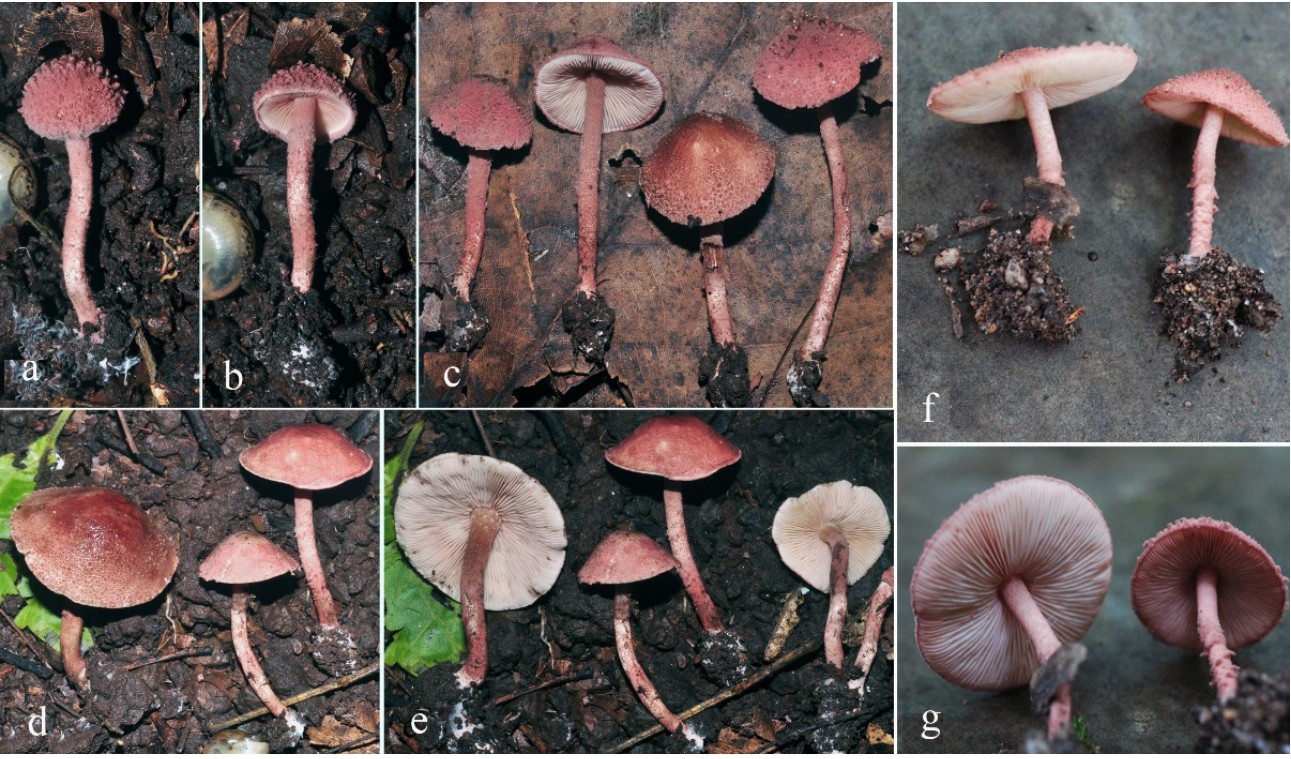

**Figure 6.** *Cystolepiota rhodella*. (**a**–**c**) = MFLU22-0019, (**d**,**e**) = HNL501799 (holotype), (**f**,**g**) = (MFLU09-0050).

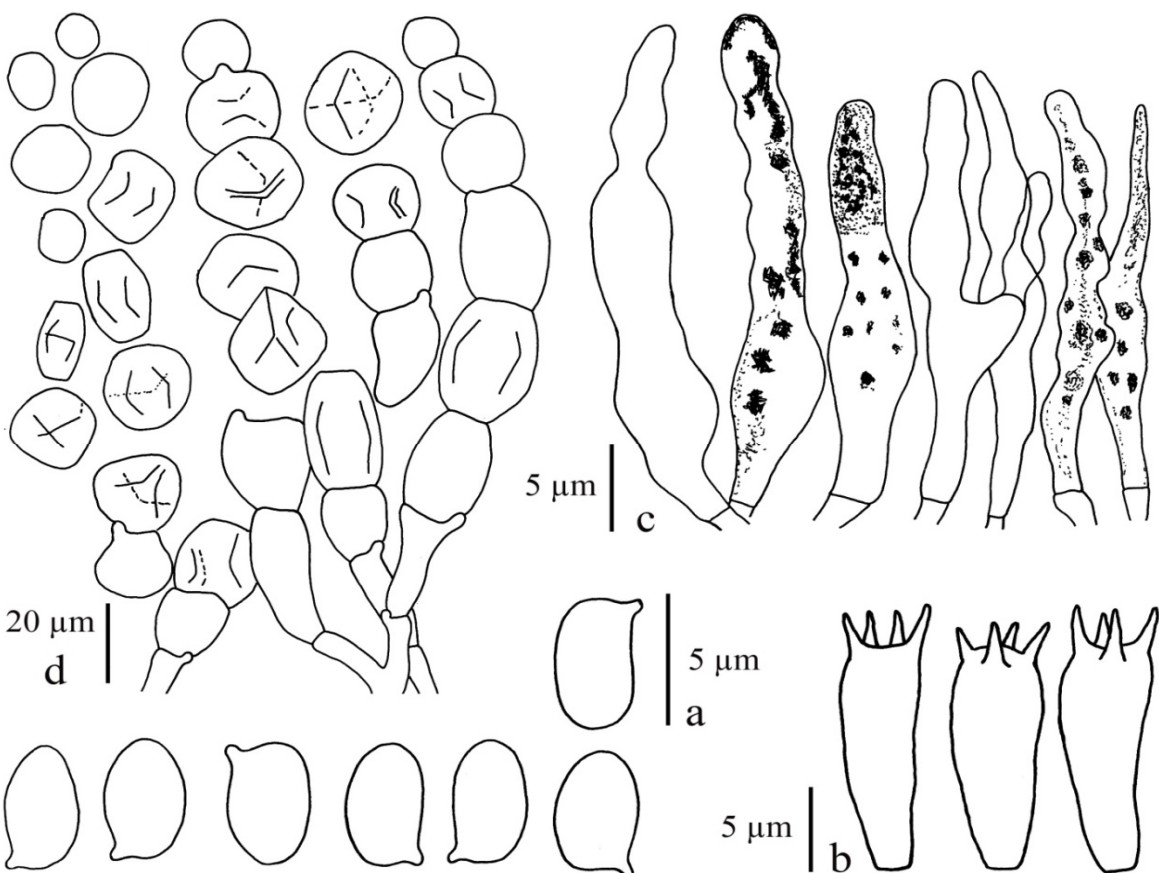

**Figure 7.** *Cystolepiota rhodella* (HNL503227, holotype) (**a**) = basidiospores, (**b**) = basidia, (**c**) = cheilocystidia, (**d**) = pileus covering elements.

**Author Contributions:** All authors have contributed equally to this work. Conceptualization, P.S. and E.C.V.; methodology, P.S., N.T. and E.C.V.; software, P.S., Y.S.L. and E.C.V.; validation, E.C.V. and P.S.; formal analysis, N.T., P.S. and E.C.V.; investigation, Y.S.L. and E.C.V.; resources, P.S. and Y.S.L.; data curation, E.C.V. and P.S.; writing—original draft preparation, P.S., E.C.V., N.T. and Y.S.L.; writing—review and editing, E.C.V. and N.T.; visualization, P.S. and N.T.; supervision, E.C.V. and N.T.; project administration, P.S. and N.T.; funding acquisition, N.T. All authors have read and agreed to the published version of the manuscript.

**Funding:** This research was funded by Thailand research fund grant "Study of saprobic Agaricales in Thailand to find new industrial mushroom products" (Grant No. DBG6180015), Thailand Science Research and Innovation (TSRI) grant "Macrofungi diversity research from the Lancang-Mekong Watershed and surrounding areas" (Grant No. DBG6280009), and Mae Fah Luang University grant "Taxonomy, phylogeny of micro and macro fungi in Mae Fah Luang University premises" (Grant No. 64316001).

**Institutional Review Board Statement:** Not applicable.

**Informed Consent Statement:** Not applicable.

**Data Availability Statement:** Data can be found within the manuscript.

**Acknowledgments:** We would like to thank the Center of Excellence in Fungal Research, Mae Fah Luang University, for providing laboratory facilities for morphological study and depositing the herbarium. The Biotechnology and Ecology Institute, Ministry of Science and Technology of Laos, is also thanked for depositing the herbarium.

**Conflicts of Interest:** The authors declare no conflict of interest.

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
