# Peer review of "Three New Species of Cystolepiota from Laos and Thailand"

_diversity, doi:10.3390/d14060449_

Round 1

Reviewer 1 Report

This study described three new species of Cystolepiota from tropical Laos and Thailand based on morphological and molecular data, and similar species were compared to the novel species. The manuscript is generally comprehensive and excellent, while it can be improved by considerations of the comments below:

  1. Sequences of Smithiomyces should be included as they are also closely related to Cystolepiota.
  2. Lack of the comparisons with some species closely related to the novel Cystolepiota rhodella. E.g., Cystolepiota rubra described from Argentina and Cystolepiota squamulosa described from China are also very similar with Cystolepiota rhodella. Comparisons should be made among these species.
  3. Some citations are incorrect or missing.
  4. Certain terms are lack of explanations.

Minor changes and minor corrections need to be made:

Line 51: Wrong citation: Bon transferred Lepiota sect. Echinatae to Cystolepiota in 1977, not in 1991. Echinoderma was proposed in Bon (1991) for original Lepiota sect. Echinatae. Add citation: Bon, M. Les lépiotes de l’herbier Boudier au Muséum National d’Histoire Naturelle de Paris. Documents Mycologiques, 1977, 7, 11-22.

Line 53: Add citation for “……in 1980 Knudsen came back from……”: Knudsen, H. A revision of Lepiota sect. Echinatae and Amyloideae (Agaricaceae) in Europe. Botanisk Tidsskrift, 1980, 75, 121-155.

Line 55: Remove the “,” of “showed, that”.

Line 96: Add the MAFFT version used.

Line 100: Add the RAxML version used.

Line 138/figure1: “New sequences generated for this study are in blue”. As shown in Figure 1, New sequences are in black and bold.

Line 168: The meaning of “avl × avw”, “Q” and “avQ” should be explained in Materials and Methods part.

Line 206: Cystolepiota oliveirae has white to cream lamellae, which may be unstable in different polulations. The other character to distinguished it from Cystolepiota pyramidalis may be the rough basidiospores.

Line 238: Spore print.

Line 252: In “By morphology, Cystolepiota thailandica is quite similar to Cystolepiota seminuda”, point out which aspects are similar.

Line 256-257: “Some of Cystolepiota seminuda has white lamellae and the stipe will discolor when touched”. This is not a good character for distinguishing the two species.

Line 257: Add a space between “discolouring” and “when”.

Line 407: The title is “Notes on Cystolepiota seminuda”.

Table1: Cystolepiota bucknallii (G0143) does not appear in the final tree of Figure1.

Author Response

Response to comments of reviewer 1

We would like to thank you very much for your kindly check in our article. We followed your comments and details of correcting are below. And the yellow marks in the manuscript are what we corrected all reviewers’ comments. 

  1. Sequences of Smithiomyces should be included as they are also closely related to Cystolepiota.

Answer: we added sequences of Smithiomyces and did new analysis. A new phylogram is provided.

  1. Some citations are incorrect or missing.

Answer: we have checked the references and corrected them.

  1. Certain terms are lack of explanations

Answer: some terms in the description, we do not explain all. We only mention that we follow the reference Vellinga & Noordeloos (2001).

Minor changes and minor corrections need to be made:

  1. Line 51: Wrong citation: Bon transferred Lepiota sect. Echinatae to Cystolepiota in 1977, not in 1991. Echinoderma was proposed in Bon (1991) for original Lepiota sect. Echinatae. Add citation: Bon, M. Les lépiotes de l’herbier Boudier au Muséum National d’Histoire Naturelle de Paris. Documents Mycologiques, 1977, 7, 11-Answer: the reference (Bon 1977) was added.

  1. Line 53: Add citation for “……in 1980 Knudsen came back from……”: Knudsen, H. A revision of LepiotaEchinatae and Amyloideae (Agaricaceae) in Europe. Botanisk Tidsskrift, 1980, 75, 121-155.

Answer: the reference Knudsen [8] was added.

  1. Line 55: Remove the “,” of “showed, that”.

Answer: “,” was removed.

  1. Line 96: Add the MAFFT version used.

Answer: MAFFT version 7.130-win32 was added.

  1. Line 100: Add the RAxML version used.

Answer: RAxML version 7.2.6 was added.

  1. Line 138/figure1: “New sequences generated for this study are in blue”. As shown in Figure 1, New sequences are in black and bold.

Answer: New sequences were changed to be in blue.

  1. ine 168: The meaning of “avl × avw”, “Q” and “avQ” should be explained in Materials and Methods part.

Answer: “avl × avw”, “Q” and “avQ” were explained in Materials and Methods.

  1. Line 206: Cystolepiota oliveiraehas white to cream lamellae, which may be unstable in different populations. The other character to distinguished it from Cystolepiota pyramidalis may be the rough basidiospores.

Answer: the colour of lamellae and rough basidiospores were compared in the note. 

  1. Line 238: Spore print

Answer: “spore pint” was changed to “spore print”.

  1. Line 252: In “By morphology, Cystolepiota thailandicais quite similar to Cystolepiota seminuda”, point out which aspects are similar.

Answer: Cystolepiota seminuda (Lasch) Bon is similar to C. thailandica by having globose and sphaero-pedunculate element cells on pileus- and stipe covering and absence of cheilo- and pleurocystidia.

  1. Line 256-257: “Some of Cystolepiotaseminuda has white lamellae and the stipe will discolor when touched”. This is not a good character for distinguishing the two species.

Answer: the paragraph was removed.

  1. Line 257: Add a space between “discolouring” and “when”.

Answer: this part was deleted.

  1. Line 407: The title is “Notes on Cystolepiota seminuda”.

Answer: we have corrected the refence.

  1. Table1: Cystolepiota bucknallii(G0143) does not appear in the final tree of Figure1.

Answer: Cystolepiota bucknallii (G0143) was removed from the Table 1.

Reviewer 2 Report

This paper needs minor revisions. This paper presents interesting information on the taxonomy of three new species of Cystolepiota. However, some points require clarification.

1.For the Cystolepiota thailandica Yuan S. Liu, Sysouph. & Thongkl. sp. nov. there is only one specimen (MFLU22-0017), normally its not enough to descript one new species.

  1. Format all the references according to Diversity journal format.
  2. Separate the long sentences since its quite hard and messy to read your manuscript. 
  3. English should be improved. 

Author Response

Response to comments of reviewer 2

We would like to thank you very much for your kindly check in all detail of our article. We followed your comments and details of correcting are below. And the yellow marks in the manuscript are what we corrected all reviewers’ comments. 

  1. For theCystolepiota thailandica Yuan S. Liu, Sysouph. & Thongkl. sp. nov. there is only one specimen (MFLU22-0017), normally it’s not enough to descript one new species.

Answer: For Cystolepiota thailandica Yuan S. Liu, Sysouph. & Thongkl. sp. nov., we have two collections (MFLU22-0017, MFLU22-0018) in the same collecting site. They were only found in northern Thailand, so we give the name as “Cystolepiota thailandica”.

  1. Format all the references according to Diversityjournal format

Answer: We have checked the format of references and some were corrected.  

  1. English should be improved.

Answer: We have checked and corrected some error English.

Reviewer 3 Report

The paper is well-written and the new species are soundly justified.

Suggested changes are minor:

line 28 - reword: . . .widespread and comprises saprotrophic species
58 - delete "these" 
71 - is "Lao" correct or should it be Laos?
147 - replace speices with species
151: clamp-connections is usually not hyphenated
173 - The term "excrescence" is not commonly used and suggests an outgrowth, when you probably mean " with a narrowed apex"(or neck), or narrowing at the apex.
222; 240 - replace ellipsoi with ellipsoid 
257 - insert space between discolouring and when 
277 - violet-brown (close up)
English vs. American spelling, for example, colour vs. color.  Perhaps the editor can decide what he/she wants for the spelling.

Author Response

Response to comments of reviewer 3

We would like to thank you very much for your kindly check in our article. We followed your comments and details of correcting are below. And the yellow marks in the manuscript are what we corrected all reviewers’ comments. 

  1. Line 28: reword “The genus Cystolepiota Singer (Agaricaceae s.l.) is widespread and comprises saprotrophic species”.

Answer: the sentence was changed to “The genus Cystolepiota Singer (Agaricaceae s.l.) is very diverse, and all species are saprotrophic.”

  1. Line 58: delete “these”

Answer:  these was deleted.

  1. Line 71: is "Lao" correct or should it be Laos?

Answer: “Lao” is nationality or adjective, but Laos is country or noun. So, we use Lao specimens.

  1. Line 147: replace speices with species

Answer: the word “species” was corrected.

  1. Line 151: clamp-connections is usually not hyphenated

Answer: “clamp-connections” was changed to be “clamp connections”

  1. Line 173: The term "excrescence" is not commonly used and suggests an outgrowth, when you probably mean " with a narrowed apex"(or neck), or narrowing at the apex.

Answer: "excrescence" was changed to “with a narrowed apex”.

  1. Line 222: replace ellipsoi with ellipsoid.

Answer: the word was changed to “ellipsoid”

  1. Line 240: replace ellipsoi with ellipsoid

Answer: the word was changed to “ellipsoid”

  1. Line 257: insert space between discolouring and when.

Answer: space was inserted.

  1. Line 277: violet-brown (close up). English vs. American spelling, for example, colour vs. color.  Perhaps the editor can decide what he/she wants for the spelling.

Answer: violet-brown.

Reviewer 4 Report

The manuscript “Three new species of Cystolepiota from Laos and Thailand” has quality and was prepared based on molecular and morphological data, that is necessary in the new taxa descriptions of the Agaricales. The following are suggestions and corrections:

Introduction

  1. Line 48 and 53: L. sistrata and E. asperum these species appear for the first time in the paper, so they need to be identified their authors
  2. Line 53: Knudsen change Kanudsen [8]

Materials and methods

  1. Line 70: Thai possibly it is an error, must be Thailand.
  2. Line 108 and 109: tree: the usual in phylogeny is phylogram.

Results na discussion

  1. Line 113, 116, 120, 124, 126, 137, 263, 313 and 329: tree: the usual in phylogeny is phylogram.
  2. Line 146: Cystolepiota pyramidalis change to C. pyramidalis, the usual in taxonomy paper, is that in the first citation of a species it should be cited with the genus written and the author of the species name, after the second the genera is abbreviated. Change at all text.
  3. Line 146 - 151: The diagnosis of a species must list the main characters of taxonomic value, without discussing them.
  4. Line 147, 221 and 276: small basidiomata and medium. This expression has no taxonomic value, because when writing scientific papers, this term must be related to something of a known size, need to compared with something known. Suggestion replace with the size that appears in the description.
  5. Line 148 and 190: pale yellow to pastel. this is not clear, as there are pastels in many shades of yellow. Suggestion quote colors in the description.
  6. Line 178: In sub-title “Habitat and distribution”: the term distribution in taxonomy is usual to refer to areas and regions where a taxa occurs. In the Agaricales taxonomy when refers to “typically scattered” or “solitary” it usual is habit. Suggestion for this is: “Habit and habitat or Habitat and habit”.
  7. Line 192 to 197: All species when cited for the first time in the text must be with the author, as was done in the following paragraph (Line 198 to 205).

References

  1. Line 408: Check this reference, [29] is not found cited in the text.

Author Response

Response to comments of reviewer 4

We would like to thank you very much for your kindly check in our article. We followed your comments and details of correcting are below. And the yellow marks in the manuscript are what we corrected all reviewers’ comments. 

     Introduction

  1. Line 48 and 57: L. sistrata and E. asperum these species appear for the first time in the paper, so they need to be identified their authors.

Answer: the authors were ere added after both species.

  1. Line 51: Knudsen change Kanudsen [8]

Answer: “Kanudsen” was changed to Knudsen

Materials and methods

  1. Line 70: Thai possibly it is an error, must be Thailand.

Answer: “Thai” is nationality or adjective, but Thailand is country or noun. So, we use Thai specimens.

  1. Line 108 and 109: tree: the usual in phylogeny is 

Answer: the word “Phylogenetic tree” was changed to be “phylogram

Results and discussion

  1. Line 113, 116, 120, 124, 126, 137, 263, 313 and 329: tree: the usual in phylogeny is phylogram.

Answer: all “Phylogenetic tree” were changed to be “phylogram”.

  1. Line 1146: Cystolepiota pyramidalischange to  pyramidalis, the usual in taxonomy paper, is that in the first citation of a species it should be cited with the genus written and the author of the species name, after the second the genera is abbreviated. Change at all text.

Answer: Cystolepiota pyramidalis change to “C. pyramidalis”.

  1. Line 146 - 151: The diagnosisof a species must list the main characters of taxonomic value, without discussing them.

 Answer: We changed the paragraph from “C. pyramidalis is distinguished from other similar Cystolepiota species by small basidiomata that are covered with light brown to brown pyramidal or irregular pyramidal squamules, pale yellow to pastel yellow lamellae, hyaline and ellipsoid-ovoid, smooth basidiospores, variably shaped cheilocystidia with or without excrescence at apex, the absence of pleurocystidia, epitheliod pileus and stipe covering, and the presence of clamp-connections” to be “C. pyramidalis is recognized by basidiomata covered with light brown to brown pyramidal or irregular pyramidal squamules, pale yellow lamellae, hyaline and ellipsoid-ovoid, smooth basidiospores, variably shaped cheilocystidia with or without excrescence at apex, the absence of pleurocystidia, epitheliod pileus and stipe covering, and the presence of clamp connections”.

  1. Line 147, 221 and 276: small basidiomata and medium. This expression has no taxonomic value, because when writing scientific papers, this term must be related to something of a known size, need to compared with something known. Suggestion replace with the size that appears in the description.

Answer: the word “small and medium” were deleted.

  1. Line 148 and 190: pale yellow to pastel. this is not clear, as there are pastels in many shades of yellow. Suggestion quote colors in the description.

Answer: we delete the word “pastel yellow” and we only keep the word _pale yellow”

  1. Line 178: In sub-title “Habitat and distribution”: the term distribution in taxonomy is usual to refer to areas and regions where a taxa occurs. In the Agaricales taxonomy when refers to “typically scattered” or “solitary” it usual is habit. Suggestion for this is: “Habit and habitat or Habitat and habit”.

 Answer: violet-brown: we change from “Habitat and distribution to be “Habitat and habit”, and also same in other species.  

  1. Line 192 to 197: All species when cited for the first time in the text must be with the author, as was done in the following paragraph (Line 198 to 205).

Answer: the authors were added after those species names.

Round 2

Reviewer 1 Report

1. certain taxon label is missing in the phylogram;

2. Some reference is not formatted to the requirements of Diversity journal. For instance, Donk (1962).

Author Response

Response to comments of reviewer 1 (round 2)

We have corrected the reviewer's comment in the yellow marks with “Track Changes” 

1. Certain taxon label is missing in the phylogram.

Answer: Before Smithiomyces lepiotoides HKAS 54390 was missed in the phylogram. Now, it has been added in the group of Smithiomyces.

2. Some reference is not formatted to the requirements of Diversity journal. For instance, Donk (1962).

Answer:

  • Line 356-357, reference number 1, "Fungal Divers." was changed to be in italic.
  • Line 366, reference number 7, "," was added after "1966".
  • Line 387-38, reference number 19, "," was added after "16".